# Mineral Monitorization in Different Tissues of *Solanum tuberosum* L. during Calcium Biofortification Process

Ana Rita F. Coelho [1,2,*], Fernando Cebola Lidon [1,2], Cláudia Campos Pessoa [1,2], Diana Daccak [1,2], Inês Carmo Luís [1,2], Ana Coelho Marques [1,2], José Cochicho Ramalho [2,3], José Manuel N. Semedo [2,4], Maria Manuela Silva [1,2], Isabel P. Pais [2,4], Maria Graça Brito [1,2], José Carlos Kullberg [1,2], Paulo Legoinha [1,2], Maria Simões [1,2], Paula Scotti-Campos [2,4], Maria Fernanda Pessoa [1,2] and Fernando Henrique Reboredo [1,2]

1  Departamento Ciências da Terra, Faculdade de Ciências e Tecnologia, Universidade NOVA de Lisboa, Campus da Caparica, 2829-516 Caparica, Portugal
2  Unidade de Geobiociências, Geoengenharias e Geotecnologias (GeoBioTec), Faculdade de Ciências e Tecnologia, Universidade NOVA de Lisboa, Campus da Caparica, 2829-516 Caparica, Portugal
3  PlantStress & Biodiversity Lab, Centro de Estudos Florestais (CEF), Instituto Superior Agronomia (ISA), Universidade de Lisboa (ULisboa), Quinta do Marquês, Av. República, 2784-505 Oeiras, and Tapada da Ajuda, 1349-017 Lisboa, Portugal
4  INIAV—Instituto Nacional de Investigação Agrária e Veterinária, Quinta do Marquês, 2784-505 Oeiras, Portugal
*  Correspondence: arf.coelho@campus.fct.unl.pt

**Abstract:** Calcium is one of the 16 essential elements for plants, being required as $Ca^{2+}$ and being involved in several fundamental processes (namely, in the stability and integrity of the cell wall, the development of plant tissue, cell division, and in stress responses). Moreover, Ca plays an important role in potato production. In this context, this study aimed to monitor the culture development (in situ and using an unmanned aerial vehicle (UAV)) and the mineral content of four essential elements (Ca, P, K, and S) in different organs of *Solanum tuberosum* L. (roots, stems, leaves, and tubers) during a calcium biofortification process, carried out with two types of solutions ($CaCl_2$ and Ca-EDTA) with two concentrations (12 and 24 kg·ha$^{-1}$). The calcium content generally increased in the *S. tuberosum* L. organs of both varieties and showed, after the last foliar application, an increase in Ca content that varied between 5.7–95.6% and 20.7–33%, for the Picasso and Agria varieties, respectively. The patterns of accumulation in both varieties during the biofortification process were different between the variety and mineral element. Regarding the quality analysis carried out during the development of the tubers, only the Agria variety was suitable for industrial processing after the last foliar application.

**Keywords:** leaves; mineral accumulation; NDVI; roots; stems; tubers



## 1. Introduction

The potato (*Solanum tuberosum* L.) tuber had its origin in the Andes (South America) and spread worldwide as an important food crop for human consumption. The potato tuber has a low protein content, although it is rich in carbohydrates (starch) and vitamin C, as well as being a good source of several B vitamins and K [1]. Due to its large genetic diversity and current demand, potato research can contribute to sustainable agrifood systems to achieve Zero Hunger and the Sustainable Development Goals [2]. The International Year of the Potato (IYP) in 2008 was a celebration of one of humanity's most important staple foods [2].

In this context, several attempts have been made in order to enrich potatoes with micro or macronutrients poorly present in tubers. When studying the biofortification of potato with iodine through the soil application of KI and the foliar application of $KIO_3$ in doses up to 2.0 kg I ha$^{-1}$, in a three-year field experiment, it was concluded that no negative

effects of iodine application on potato yield or dry matter content were observed [3]. Both applications allowed to obtain potato tubers with increased content of iodine without a decrease in the starch or sugar content. The highest efficiency was noted for foliar spraying with $KIO_3$ in a dose of 2.0 kg I ha$^{-1}$. The obtained level of iodine in 100 g of potatoes could be sufficient to cover up to 25% of Recommended Daily Allowance for the element [3].

Andean potato cultivars can be Zn-biofortified through the foliar and soil application of Zn- fertilizers. High rates of foliar application reached a 2.51-fold tuber Zn increase, while high rates of soil application had a 1.91-fold tuber Zn increase in the field trials. Conversely, tuber Fe concentrations of the same Andean cultivars were not increased with Fe fertilization [4]. The artificial enrichment of crops may lead to interactions among nutrients which is well documented in various studies of plant physiology [5–8]; thus, when promoting biofortification, it is important to evaluate putative imbalances in nutrient composition.

For example, both the foliar and soil application of iodine under the form of $KIO_3$ (fo-liar) and KI (soil) contributed to higher contents of K, Mg, Ca, Mn, and Cd in lettuce, as well as to a decrease in the P, Cu, and Zn levels, compared to control plants. Doses, forms, and application methods of iodine are responsible for the diversity of effects on the elemental content of Al, B, Fe, Mo, Na, S, and Pb in lettuce plants [9]. Another study with lettuce showed that the combined biofortification of soilless-grown lettuce with I, Se, and Zn at the rates of 150, 20, and 50 mM, respectively, in nutrient solution, showed that combined biofortification with I–Se–Zn had no significant effect on the concentrations of most of the essential elements to plants, although, for example, S and Si had increased, and Fe and Mn had decreased [10].

In the same context, the excess of Zn through biofortification can negatively influence the uptake of Mn and Fe [11], whereas the Ca levels of wheat and sunflower cultivars were reduced by P fertilization while increasing in chickpea and lentil. In general, the Fe concentrations of the wheat and chickpea cultivars were significantly increased by P, while the Zn and Cu concentrations of all the cultivars of the four species were reduced, particularly Zn [12]. Thus, however the agronomic biofortification scheme was applied in potato research, the goal must take into account not only the yield but also the nutritional value of the tuber.

Beyond the best choice of cultivar, adequate nutrient management, where water supply is included, and climate are the main factors of potato production. Regarding climate and taking into account the origin of potato (the high Andean plateau), tubers from those locales are well-adapted to strong UV-B radiation as well as other native flora [13], and current climate changes will pose a threat to potato cultivars from low altitudes that can be forced to migrate upwards, thus it will be important in the near future to develop tolerant UV-B cultivars [14,15]. Additionally, the drought periods will be intensified, as well as $CO_2$ concentrations, which are climbing [16]. Additionally, a sufficient supply of macro and micronutrients is crucial for achieving high yield and for producing potatoes with nutritional quality. The nutrients which are most commonly fertilized in potato production are N, P, and K [17], although the rates of N, P, and K fertilizers used in potato cultures are often considered excessive, as they can cause imbalance among the essential elements present in the soil [18].

Potassium is found in large quantities in potato plants, being essential in starch transformation and movement from leaves to tubers, and in carbohydrate formation [19]. Additionally, the use of K fertilizer improved tuber yield and the number of marketable tubers in all planting patterns and irrigation levels, especially in plants with hydric stress— submitted to only 60% of their water requirement [20].

Potato crops possess a high requirement for the available P in soil. Once entering via the root system, P participates in various metabolic processes including energy transfer, the synthesis of nucleic acids and starch, the synthesis and stability of membranes, the activation of enzymes, respiration, redox reactions, and carbohydrate metabolism [21,22]. Phosphorus is considered one of the more mobile nutrients in plants, despite the low P uptake efficiency in potato, mainly attributed to a relatively low root-to-shoot ratio

and especially to a relatively low proportion of root hairs [19,23]. When evaluating the chemical composition of potato tubers (five cultivars) grown in soils with low, medium, and high availability of P, it was concluded that the increased availability in soil allowed the production of tubers with higher dry matter content, lower total sugar content, and a higher percentage of starch and protein [24].

Sulfur is a structural component of protein disulfide bonds, the formation of sulfur amino acids, vitamins, and cofactors. Sulfur and sulfur-containing compounds act as signaling molecules in stress conditions [25]. Sulfur application in four varieties of potato showed a significant influence on the quality and yield. These parameters increased with increasing doses of sulfur up to 45 kg·ha$^{-1}$, while beyond that level, no significant improvements were noted [26]. A three-year research study of potato fertilization with 20 and 40 kg·ha$^{-1}$ as $K_2SO_4$, $(NH_4)SO_4$, and elemental S revealed a significant increase in the tuber yield and the protein content compared to the control. However, no significant effect of fertilization on the content of dry matter and starch in potato tubers was observed. Regardless of the form of S used, no clear difference in the potato tuber yield, size, and protein was noted [27].

Calcium is an essential element for plants, being important for cell wall and membrane stability, and as a second messenger in many physiological processes, including the response of plants to biotic stress [28,29]. Using a substrate containing 50% perlite–50% sand, it was possible to grow potato plants with different combined concentrations of Ca and P [30]. It was observed that a solution containing 42 mg·kg$^{-1}$ P and 120 mg·kg$^{-1}$ Ca is sufficient to maximize the potato tuber number. A nutrient solution containing 21 mg·kg$^{-1}$ P and 120 mg·kg$^{-1}$ Ca is the best treatment to maximize the total tuber yield [30].

The aim of the present work is to conduct a Ca biofortification process, carried out with $CaCl_2$ and Ca-EDTA, at two different concentrations (12 and 24 kg·ha$^{-1}$) on two varieties of potato widely cultivated in Portugal (Picasso and Agria) in field conditions. Potato plant development was monitored through UAV and in situ during the biofortification process, in order to identify signals of toxicity. Elemental analyses were carried out after the fourth, sixth, and seventh foliar applications in order to study the elemental ac-cumulation (P, K, S, and Ca) in the roots, stems, leaves, and tubers of *S. tuberosum* L. and identify the patterns of the accumulation and possible interference of Ca fertilization on the uptake and translocation of the above-mentioned elements. Additionally, during the biofortification process, quality analyses were performed in the tubers.

## 2. Materials and Methods

### 2.1. Biofortification Itinerary

The two experimental fields used to grow two commercial varieties of *S. tuberosum* L. (Picasso and Agria), were in the same region of Western Portugal (Lourinhã) (Figure 1). After the beginning of the tuberization, seven foliar applications were carried out on the same date for both varieties with two concentrations (12 and 24 kg·ha$^{-1}$) of $CaCl_2$ (pH 4–6) or, alternatively, Ca-EDTA (pH 7). The planting dates were on 21 March 2019 and 15 March 2019 for Picasso and Agria, respectively. Foliar applications were carried out on 30 May, 7 June, 14 June, 21 June, 28 June, 4 July, and 12 July 2019. Harvesting occurred on 9 August and 29 July 2019 for Picasso and Agria, respectively. Control plants were not sprayed at any time with $CaCl_2$ or Ca-EDTA. Considering that Ca-EDTA might become highly toxic to plants, only one foliar application was carried out with Ca-EDTA 24 kg ha$^{-1}$; the remaining treatments were performed seven times/foliar applications. During the plant cycle and agricultural period (15 March to 9 August 2019), the air temperatures ranged between 13.8 and 21.9 °C, and the average rainfall was 0.51 mm.

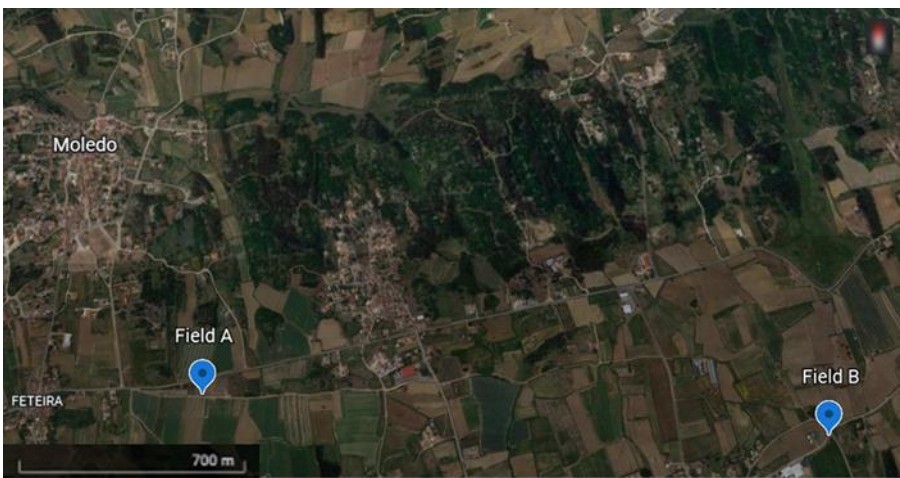

**Figure 1.** Geographic location of the two experimental fields (Field A and B) (images obtained through Google Earth). Field A—GPS coordinates: 39°16′39′′ N 9°15′08′′ W (142 m) and Field B—GPS coordinates: 39°16′31′′ N 9°13′47′′ W (182 m).

### 2.2. Ortophotomaps and NDVI (Normalized Difference Vegetation Index) Models in the Experimental Fields

The ortophotomaps and the NDVI model (varying between −1 to 1) of both fields were processed in ArcGIS Pro after the images were obtained with a UAV (unmanned aerial vehicle) equipped with altimetric measurement sensors and synchronized by GPS. The flight was performed on 25 June (four days after the fourth foliar application) and 10 July (six days after the sixth foliar application) in Fields A and B, respectively, to characterize the vegetation indexes, to monitor differences in the vigor between the control and the plants sprayed with two concentrations (12 and 24 kg·ha$^{-1}$) of $CaCl_2$ or, alternatively, Ca-EDTA.

### 2.3. Mineral Content in Roots, Stems, Leaves, and Tubers

The mineral content of Ca, K, S, and P, were determined in the roots, stems, leaves, and tubers (being cut, dried at 60 °C until constant weight, and grounded) after the fourth, sixth, and seventh foliar applications with $CaCl_2$ or, alternatively, Ca-EDTA in two concentrations (12 and 24 kg ha$^{-1}$) using an XRF analyzer (model XL3t 950 He GOLDD+) under a He atmosphere, according to [31]. The analyses were carried out in quadruplicate (using different plants) per treatment.

### 2.4. Quality Parameters

The height (cm), diameter (cm), and dry weight were measured considering four randomized tubers from different plants per treatment after the fourth, sixth, and seventh foliar applications. The dry weight was measured according to [32].

### 2.5. Statistical Analysis

Statistical analysis was carried out with the IBM SPSS software using one-way and two-way ANOVA to assess the differences between the treatments in each variety (Picasso and Agria) and between both varieties in the same treatment in each parameter analyzed, followed by Tukey's analysis for mean comparison. A 95% confidence level was adopted for all of the tests. All of the statistical analysis was carried out with quadruplicates of each sample.

Additionally, Pearson's correlation was carried out after the last foliar application in both varieties and also through IBM SPSS software.

## 3. Results

### 3.1. Monitorization of Solanum tuberosum L. Plants

Orthophotomaps (Figure 2A,C) and NDVI models (Figure 2B,D) of both fields (A and B) were carried out to monitor the growth and vegetative development of *S. tuberosum* L. plants biofortified with calcium. In Field A, after the fourth foliar application, it is possible to identify a lower vigor (lower NDVI) in the orthophoto map (Figure 2A) and NDVI model (Figure 2B), corresponding to 12 kg·ha$^{-1}$ Ca-EDTA (12B) treatment. Regarding Field B (Figure 2D), after the sixth foliar application (the plants being in a more advanced stage of development), it is possible to identify three field lines most affected (in black) that show lower vigor, corresponding to 12 kg·ha$^{-1}$ Ca-EDTA, 24 kg·ha$^{-1}$ CaCl$_2$ (24A) and 12 kg·ha$^{-1}$ CaCl$_2$ (12A), respectively, from left to right.

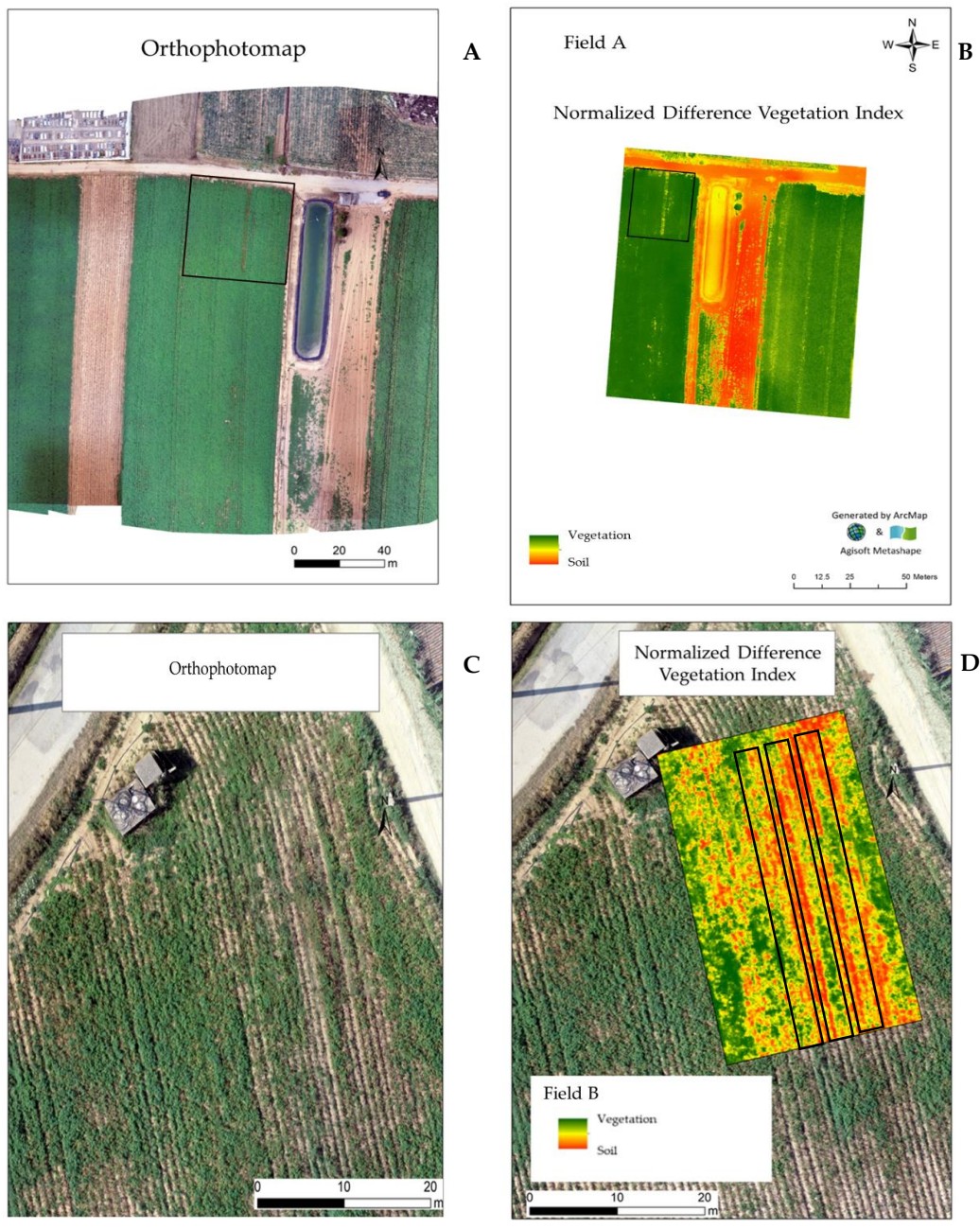

**Figure 2.** Orthophotomaps (**A**,**C**) and NDVI models (**B**,**D**) of *S. tuberosum* L. fields (Fields A—Picasso variety and B—Agria variety), after four and six foliar applications, respectively. In Field A, the information was collected four days after the fourth foliar application, and in Field B, six days after

the sixth foliar application. The indication (in black) of field limits in the maps. Regarding the NDVI models, varying from −1 to 1, the color red corresponds to −1 (soil), and the color green corresponds to 1 (higher plant vigor).

The monitorization in situ of *S. tuberosum* L. development occurrences in both fields (Figures 3 and 4). Regarding Field A (Figure 3), after six foliar applications, the Picasso variety showed major signals of negative effects in the 12B treatment, despite some minor signals of negative effects in both concentrations of CaCl$_2$. Additionally, 24 kg·ha$^{-1}$ Ca-EDTA (24B) and control plants have a very similar appearance.

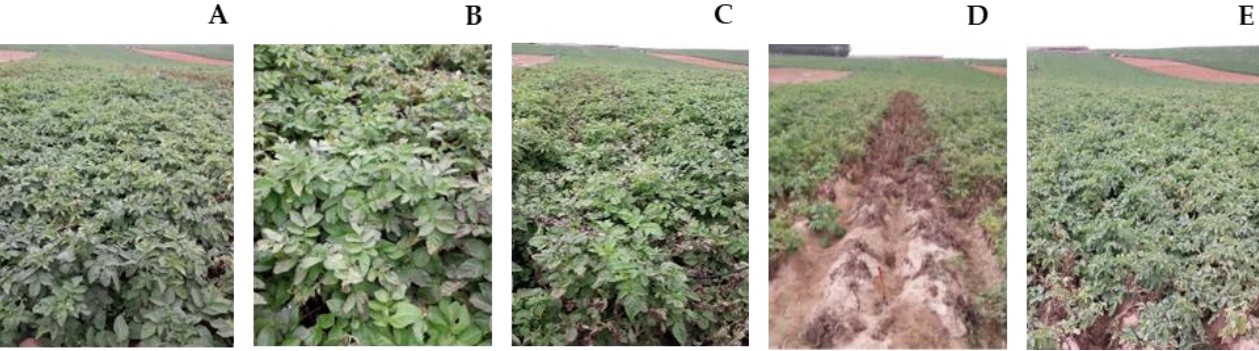

**Figure 3.** Production overview of *S. tuberosum* L. in Field A (Picasso variety) after six foliar applications: (**A**)—Control; (**B**)—CaCl$_2$ 12 kg·ha$^{-1}$ (12A); (**C**)—CaCl$_2$ 24 kg·ha$^{-1}$ (24A); (**D**)—Ca-EDTA 12 kg·ha$^{-1}$ (12B); (**E**)—Ca-EDTA 24 kg·ha$^{-1}$ (24B).

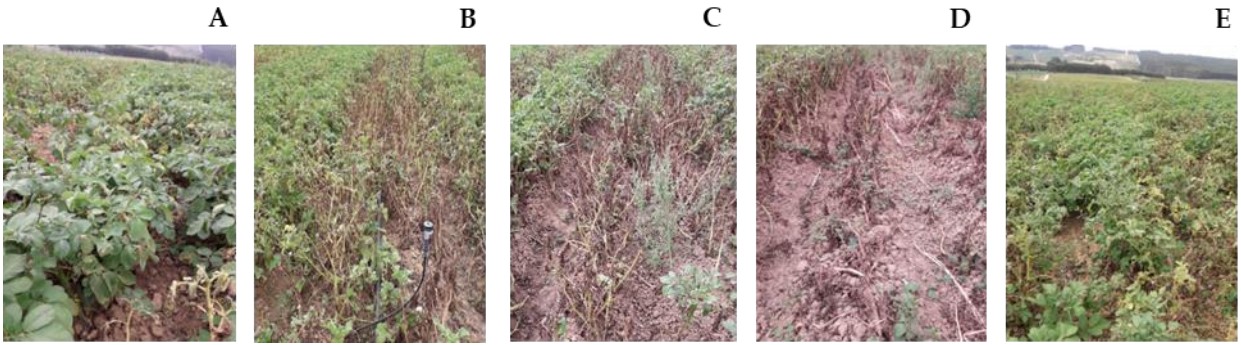

**Figure 4.** Production overview of *S. tuberosum* L. in Field B (Agria variety) after six foliar applications: (**A**)—Control; (**B**)—CaCl$_2$ 12 kg·ha$^{-1}$ (12A); (**C**)—CaCl$_2$ 24 kg·ha$^{-1}$ (24A); (**D**)—Ca-EDTA 12 kg·ha$^{-1}$ (12B); (**E**)—Ca-EDTA 24 kg·ha$^{-1}$ (24B).

In Field B (Figure 4), after six foliar applications, the Agria variety showed signals of toxicity in both concentrations of CaCl$_2$ (having more damage in the highest concentration applied) and in the 12B treatment (toxicity was more evident).

### 3.2. Mineral Content of Ca, K, S, and P after the Fourth, Sixth, and Seventh Foliar Applications

To monitor the mineral content in the different organs of *S. tuberosum* L., the roots (Figure 5), stems (Figure 6), leaves (Figure 7), and tubers (Figure 8) were analyzed after the fourth (4FA), sixth (6FA), and seventh (7FA) foliar applications. The content of the minerals analyzed (Ca, K, S, and P) showed different patterns of accumulation during the development of the plants and the different contents between the Picasso and Agria varieties after the same foliar applications.

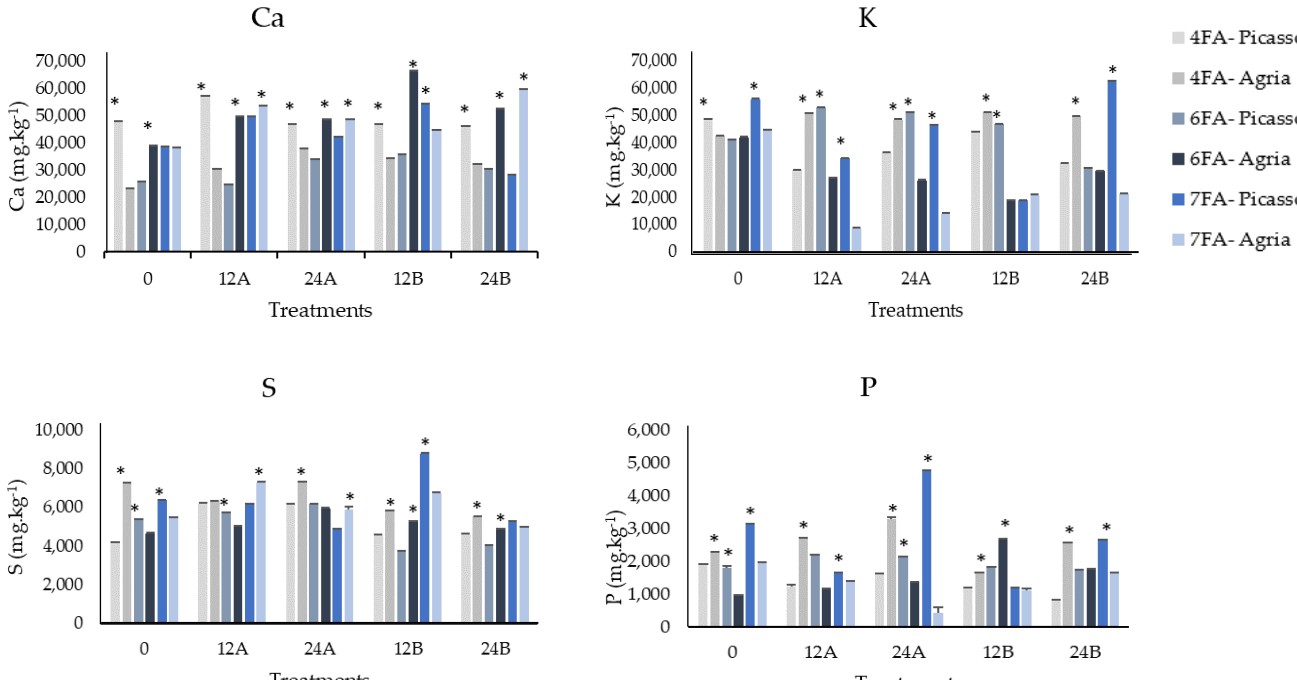

**Figure 5.** Chemical elements in roots of *S. tuberosum* L. Agria and Picasso varieties, during the production cycle of plants, after the fourth, sixth, and seventh foliar applications with $CaCl_2$ and Ca(EDTA). Foliar spray was carried out with two concentrations of $CaCl_2$ (12 (corresponding to 12A) and 24 (corresponding to 12B) kg·ha$^{-1}$) and two concentrations of Ca-EDTA (12 (corresponding to 12B) and 24 (corresponding to 24B) kg·ha$^{-1}$). The control was not sprayed at any time and corresponds to "0". Chemical quantification was carried out after the fourth (corresponding to 4FA), sixth (corresponding to 6FA), and seventh (corresponding to 7FA) foliar applications. Mean values ± S.E. ($n = 4$) and significantly higher content between varieties within each treatment after the 4FA, 6FA, and 7FA are identified (*).

Regarding the roots of *S. tuberosum* L. (Figure 5), the Ca content after the 4FA was higher with the application of $CaCl_2$ in both varieties (in the Picasso variety with the 12A treatment and in the Agria with the 24A treatment). However, after the 6FA and 7FA, a higher content was obtained with Ca-EDTA in both varieties. Additionally, comparing both varieties in the 24B treatment, the Picasso variety showed an increase in Ca with the increase in foliar applications, and the Agria variety showed a contracting pattern. Nevertheless, the control roots in the Agria variety showed the lowest content of Ca after the 4FA, 6FA, and 7FA.

Regarding K and P, there is not a clear tendency regarding the accumulation of both mineral elements in both varieties. Yet, considering K and P after the 7FA, the highest content obtained in the Agria variety was in control. Regarding K, there was a decrease in the 12A, 24A, and 24B with an increase in foliar applications in the Agria variety and a decrease in 12A, 24A, and 12B from the 6FA to the 7FA in the Picasso variety. In sulfur, after the 4FA and 6FA, both varieties showed the highest levels with $CaCl_2$. However, after the 7FA, the highest content was obtained with the concentration of 12 kg·ha$^{-1}$ with Ca-EDTA and $CaCl_2$, in Picasso and Agria, respectively. Additionally, in both varieties, in the 12A, 12B, and 24B treatments and the control (in Agria), there was an increase from the 6FA to the 7FA.

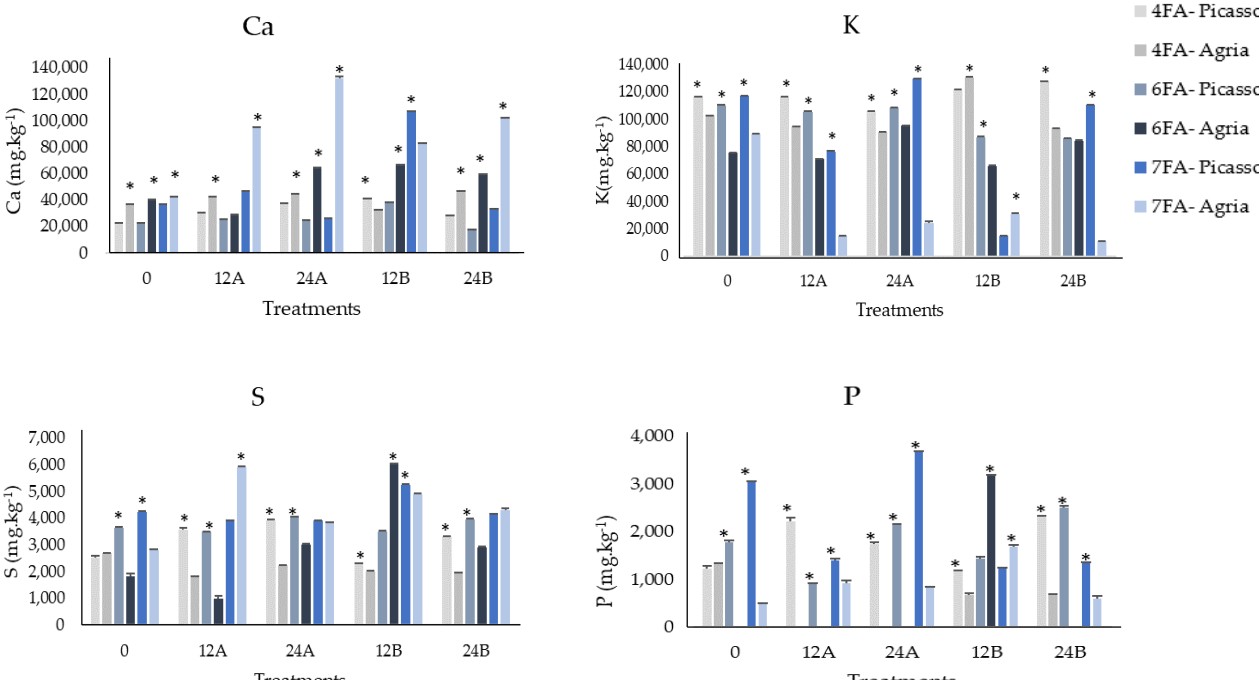

**Figure 6.** Chemical elements in stems of *S. tuberosum* L. Agria and Picasso varieties, during the production cycle of plants, after the fourth, sixth, and seventh foliar applications with CaCl$_2$ and Ca(EDTA). Foliar spray was carried out with two concentrations of CaCl$_2$ (12 (corresponding to 12A) and 24 (corresponding to 12B) kg·ha$^{-1}$) and two concentrations of Ca-EDTA (12 (corresponding to 12B) and 24 (corresponding to 24B) kg·ha$^{-1}$). The control was not sprayed at any time and corresponds to "0". Chemical quantification was carried out after the fourth (corresponding to 4FA), sixth (corresponding to 6FA), and seventh (corresponding to 7FA) foliar applications. Mean values ± S.E. (*n* = 4) and significantly higher content between varieties within each treatment after the 4FA, 6FA, and 7FA are identified (*). Values not shown in P are below the detection limit of the device (450 mg·kg$^{-1}$).

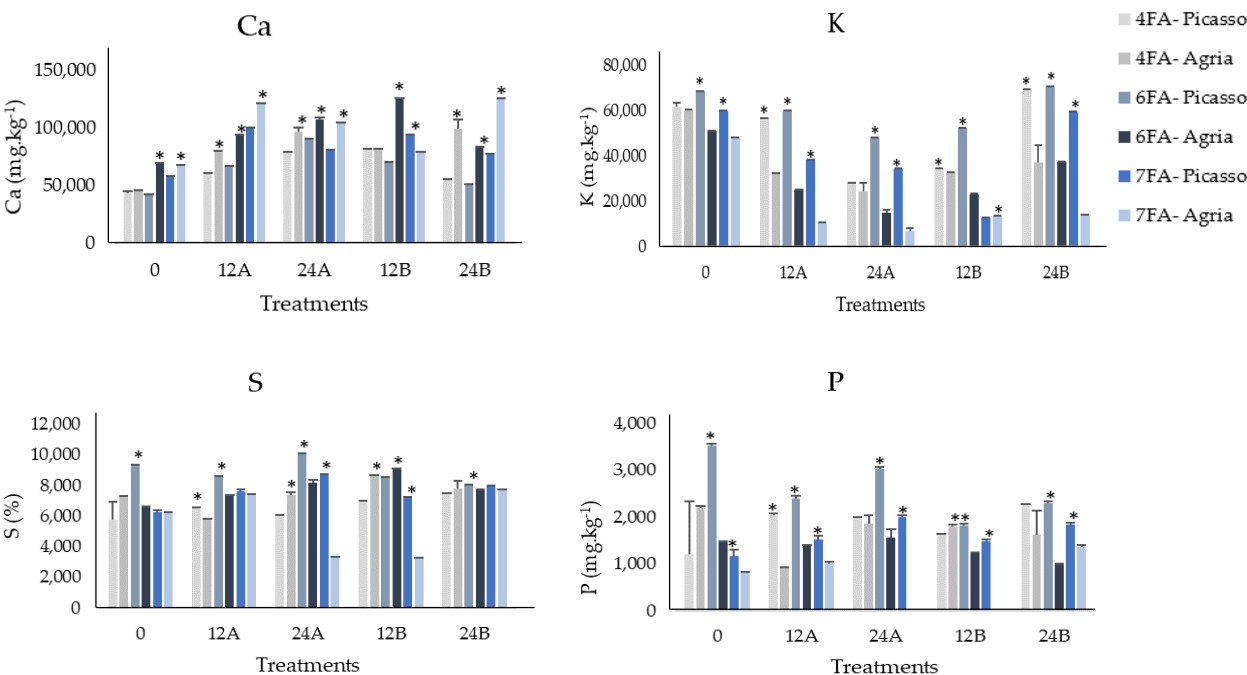

**Figure 7.** Chemical elements in leaves of *S. tuberosum* L. Agria and Picasso varieties, during the production

cycle of plants, after the fourth, sixth, and seventh foliar applications with $CaCl_2$ and Ca(EDTA). Foliar spray was carried out with two concentrations of $CaCl_2$ (12 (corresponding to 12A) and 24 (corresponding to 12B) kg·ha$^{-1}$) and two concentrations of Ca-EDTA (12 (corresponding to 12B) and 24 (corresponding to 24B) kg·ha$^{-1}$). The control was not sprayed at any time and corresponds to "0". Chemical quantification was carried out after the fourth (corresponding to 4FA), sixth (corresponding to 6FA), and seventh (corresponding to 7FA) foliar applications. Mean values $\pm$ S.E. ($n$ = 4) and significantly higher content between varieties within each treatment after the 4FA, 6FA, and 7FA are identified (*). Values not shown in P are below the detection limit of the device (450 mg·kg$^{-1}$).

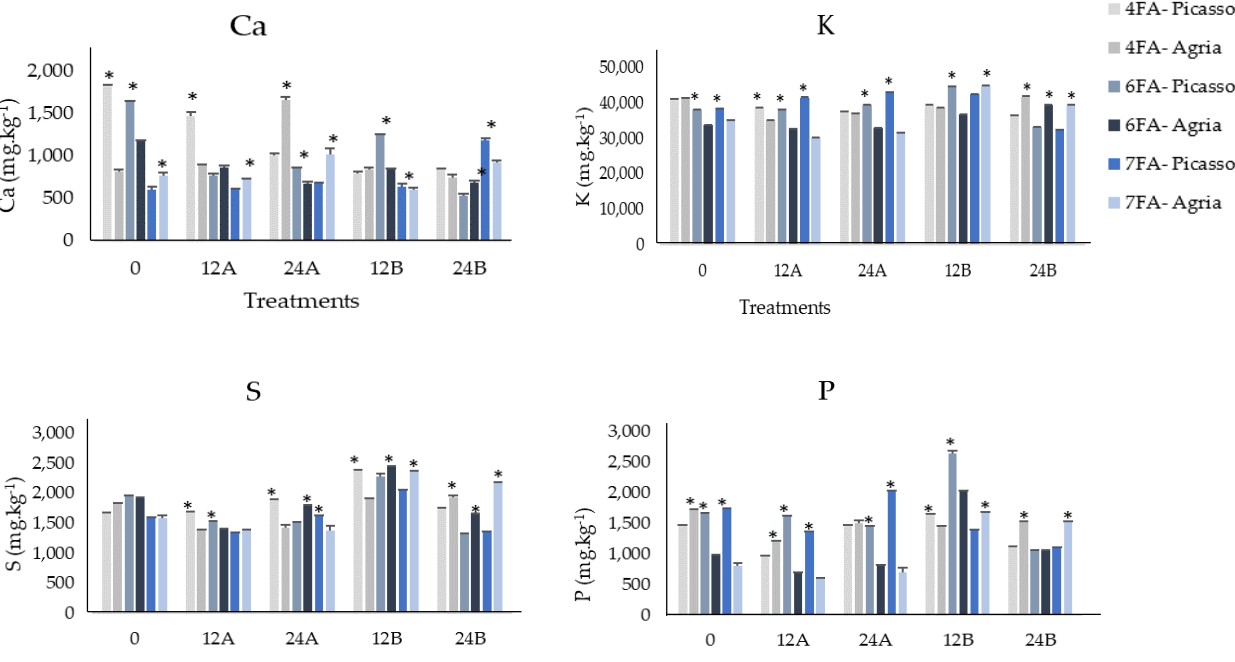

**Figure 8.** Chemical elements in tubers of *S. tuberosum* L. Agria and Picasso varieties, during the production cycle of plants, after the fourth, sixth, and seventh foliar application with $CaCl_2$ and Ca(EDTA). Foliar spray was carried out with two concentrations of $CaCl_2$ (12 (corresponding to 12A) and 24 (corresponding to 12B) kg·ha$^{-1}$) and two concentrations of Ca-EDTA (12 (corresponding to 12B) and 24 (corresponding to 24B) kg·ha$^{-1}$). The control was not sprayed at any time and corresponds to "0". Chemical quantification was carried out after the fourth (corresponding to 4FA), sixth (corresponding to 6FA), and seventh (corresponding to 7FA) foliar applications. Mean values $\pm$ S.E. ($n$ = 4) and significantly higher content between varieties within each treatment after the 4FA, 6FA, and 7FA are identified (*).

Moreover, in the Picasso variety after the 7FA, the 12B treatment showed the highest content of Ca and S but the lowest content of K and P. Additionally, treatment 24B showed the lowest content of Ca and S and the highest of K. Regarding the Agria variety after the 7FA, the 24B treatment showed the highest content of Ca and the lowest of S, and the control showed the lowest content of Ca and the highest of K and P.

Regarding the control, after the 7FA, the Picasso and Agria varieties showed an increase in Ca content that varied between 9.7–41.1% and 16.5–55.1%, respectively.

Considering the stems of *S. tuberosum* L. (Figure 6), the Ca content was higher in the 12B treatment after the 4FA, 6FA, and 7FA in the Picasso variety, and in the Agria variety, in 24A after the 7FA. Additionally, the Ca content increased with the increase in foliar applications in the Picasso variety in all the treatments applied (except in 12A). As seen in the roots, in the K content, there is not a clear tendency regarding the accumulation of both mineral elements in both varieties. Yet, after the 7FA, the highest content was obtained in 24A and the control in Picasso and Agria, respectively. Regarding S and P, in



the Picasso variety, the highest content was obtained after the 4FA and 6FA in the 24A and 24B treatments, respectively. After the 6FA, Agria showed the highest content of S and P in the 12B treatment, and after the 7FA, the highest content of S was obtained in 12 kg·ha$^{-1}$ with Ca-EDTA and CaCl$_2$, respectively, for Picasso and Agria. In treatments 24A and 12B, Picasso and Agria showed the highest content for P after the 7FA. Nevertheless, there was an increase in the S content with the increase in foliar applications in the control, 24A, 12B in Picasso, and 24B in both varieties; in P, the same occurred in the control in Picasso and in 24A in both varieties.

Moreover, as seen in the roots, in the Picasso variety after the 7FA, treatment 12B showed the highest content of Ca and S but the lowest content of K and P. In treatment 24A, the content of Ca was the lowest, and the K and P contents were the highest. Regarding the Agria variety, the control showed the lowest content of Ca, S, and P and the highest of K.

Regarding the control, after the 7FA, the Picasso and Agria varieties showed an increase in Ca content that varied between 28.2% and 3-fold and 2- to 3.1-fold, respectively.

Considering the leaves of *S. tuberosum* L. (Figure 7), the Ca content was higher in the 24B treatment after the 4FA and 7FA in the Agria variety and higher in 12A after the 7FA in the Picasso variety. Nevertheless, in both varieties, the Ca content increased with the increase in foliar applications only in 12A. Regarding K, the highest content was obtained after the 7FA was in control for both varieties. Additionally, there was a decrease in the K content with the increase in foliar applications in all the treatments (except 24B) in the Agria variety, and the content decreased in all treatments from the 6FA to the 7FA in the Picasso variety. Considering the S and P contents, after the 7FA, the highest content was obtained with the concentration of 24 kg·ha$^{-1}$ with CaCl$_2$ (in Picasso) and Ca-EDTA (in Agria). The Sulfur and P contents decreased in all treatments from the 6FA to the 7FA in both varieties (except in 12A and 24B in Agria—the content of S does not change). Furthermore, in the Agria variety after the 7FA, the control showed the lowest content of Ca, S, and P and the highest of K. Additionally, 12B showed the highest contents of Ca and S and the lowest in K and P after the 7FA, regarding the Picasso variety.

Regarding the control, after the 7FA, the Picasso and Agria varieties showed an increase in the Ca content that varied between 32.8–73.1% and 16.8–84.3%, respectively.

Considering the tubers of *S. tuberosum* L. (Figure 8), after the last foliar application, the Ca content was higher when the concentration applied was 24 kg·ha$^{-1}$ (24B in Picasso and 24A in Agria variety). Yet, the highest content of Ca was in the control in both varieties. However, in the control tubers from Picasso, there was a decrease in the Ca content (12A and 24A) with the increase in foliar applications. Regarding the K and P contents, in the Picasso and Agria varieties after the 7FA, the highest contents were obtained in 24A and 12B, respectively. In K, with the increase in foliar applications, in 12A and 24A, there is a decrease in the content in Picasso and an increase in the Agria variety. Additionally, both varieties showed a decrease in the K content with the increase in foliar applications in the 24B treatment, and in S, the 12B treatment showed a higher content compared to all the remaining treatments. As in S, P showed a higher content after the 6FA and 7FA in the 12B treatment in the Picasso variety. Generally, the P content in Picasso was higher compared to the Agria variety in each foliar application analyzed.

Moreover, in the Picasso variety after the 7FA, the 24B treatment showed the highest content of Ca and the lowest of K and P. In the Agria variety, after the 7FA, 12B treatment showed the lowest content of Ca and the highest of K, S, and P. Additionally, it was in the 12A treatment that the lowest contents of K, S, and P were obtained in Agria, after the 4FA, 6FA, and 7FA.

Regarding the control, after the 7FA, the Picasso and Agria varieties showed an increase in Ca content that varied between 5.7–95.6% and 20.7–33%, respectively. The 12A treatment in both varieties and the 12B treatment in the Agria variety were not biofortified after the 7FA.

Overall, it is possible to observe a decrease in the Ca content from the areal part of the plants to the tubers (leaves > stems > roots > tubers) in the Picasso variety (in 24B after

the 4FA, in 12A and 12B after the 6FA, and in 12B and 24B after the 7FA) and in the Agria variety in all the treatments during the biofortification monitorization (except in 12B after the 4FA and 12A after the 6FA).

Considering the height, diameter, and dry weight (Table 1) there were no significant differences between the treatments within each variety (except in height in the Agria variety after the 4FA). Yet, there is no pattern in the parameters analyzed in the Picasso variety after the 4FA and the 6FA, but after the 7FA, the control showed a larger height, diameter, and dry weight compared to the remaining treatments. After the 7FA, the height, diameter, and dry weight in the Picasso variety varied between 7.90–13.6 cm, 4.40–6.30 cm, and 15.22–17.13%, respectively. The diameter and dry weight after the 4FA in the Agria variety showed no significant differences between treatments, contrary to height. After the 6FA, the control showed a smaller height and diameter compared to the remaining treatments, and after the 7FA, the 12A treatment (as seen after the 4FA) showed a larger height and diameter. Additionally, after the 7FA, 24B showed the smallest height and diameter in this variety. Regarding dry weight, there is not a clear tendency; however, after the 7FA, the highest content was obtained in the control and the lowest in 24A. In the Agria variety, after the 7FA, the height, diameter, and dry weight varied between, 5.20–10.43 cm, 3.80–7.67 cm, and 20.83–23.67%, respectively.

**Table 1.** Height, diameter, and dry weight of the tubers of the *S. tuberosum* L. Picasso and Agria varieties after the fourth, sixth, and seventh foliar application with $CaCl_2$ (12 and 24 kg·ha$^{-1}$) and Ca-EDTA (12 and 24 kg·ha$^{-1}$). For each parameter and foliar application, the different letters after the mean values ± S.E. (*n* = 4) express significant differences between the treatments within each variety (a,b). Significant higher content between varieties within each treatment after the fourth (4FA), sixth (6FA), and seventh (7FA) foliar application is identified (*).

| Foliar Application | Treatments | Picasso | | | Agria | | |
|---|---|---|---|---|---|---|---|
| | | Height (cm) | Diameter (cm) | Dry Weight (%) | Height (cm) | Diameter (cm) | Dry Weight (%) |
| 4° | Control | 8.30 ± 2.08a * | 3.53 ± 0.78a | 18.40 ± 1.11a | 2.33 ± 0.88b | 3.00 ± 1.15a | 16.75 ± 2.40a |
| | $CaCl_2$ 12 kg·ha$^{-1}$ | 9.27 ± 1.49a | 6.17 ± 1.17a | 15.90 ± 0.70a | 8.73 ± 0.22a | 6.03 ± 0.37a | 21.24 ± 0.76a * |
| | $CaCl_2$ 24 kg·ha$^{-1}$ | 7.00 ± 0.65a * | 7.30 ± 1.15a * | 15.08 ± 0.98a | 4.80 ± 1.75ab | 3.70 ± 1.26a | 16.90 ± 3.10a |
| | Ca-EDTA 12 kg·ha$^{-1}$ | 5.20 ± 1.30a | 5.40 ± 0.57a * | 18.04 ± 2.45a | 4.17 ± 0.46ab | 3.47 ± 0.29a | 18.86 ± 0.34a |
| | Ca-EDTA 24 kg·ha$^{-1}$ | 6.80 ± 0.40a * | 4.53 ± 1.12a | 14.98 ± 1.57a | 4.63 ± 0.88ab | 3.53 ± 0.78a | 26.46 ± 10.5a |
| 6° | Control | 6.63 ± 2.19a | 4.67 ± 1.20a | 14.86 ± 1.08a | 4.77 ± 0.39a | 3.67 ± 0.18a | 21.47 ± 0.63a * |
| | $CaCl_2$ 12 kg·ha$^{-1}$ | 9.20 ± 1.68a | 7.03 ± 1.21a | 16.61 ± 0.67a | 7.50 ± 1.00a | 6.00 ± 0.40a | 23.13 ± 0.49a * |
| | $CaCl_2$ 24 kg·ha$^{-1}$ | 6.57 ± 2.13a | 4.73 ± 1.28a | 14.47 ± 1.62a | 7.77 ± 0.38a | 5.73 ± 0.20a | 28.99 ± 5.58a * |
| | Ca-EDTA 12 kg·ha$^{-1}$ | 6.90 ± 0.21a | 5.73 ± 0.52a | 16.02 ± 0.20a | 7.00 ± 2.17a | 5.53 ± 0.69a | 19.32 ± 0.34a * |
| | Ca-EDTA 24 kg·ha$^{-1}$ | 9.07 ± 2.17a | 5.40 ± 1.07a | 15.38 ± 0.30a | 6.37 ± 0.71a | 4.87 ± 0.69a | 18.08 ± 3.10a |
| 7° | Control | 13.6 ± 0.60a * | 6.30 ± 0.80a | 17.13 ± 0.93a | 7.43 ± 0.55a | 5.40 ± 0.53a | 23.67 ± 0.43a * |
| | $CaCl_2$ 12 kg·ha$^{-1}$ | 8.70 ± 1.11a | 4.40 ± 0.31a | 16.57 ± 0.78a | 10.43 ± 0.09a * | 7.67 ± 0.12a | 23.13 ± 0.78a * |
| | $CaCl_2$ 24 kg·ha$^{-1}$ | 9.27 ± 0.37a * | 5.50 ± 0.52a | 16.21 ± 1.04a | 7.53 ± 1.06a | 5.83 ± 0.77a | 20.83 ± 0.56a * |
| | Ca-EDTA 12 kg·ha$^{-1}$ | 7.90 ± 0.61a | 5.10 ± 0.21a | 15.84 ± 0.36a | 7.20 ± 1.90a | 6.40 ± 1.40a | 23.46 ± 0.19a * |
| | Ca-EDTA 24 kg·ha$^{-1}$ | 8.20 ± 1.52a * | 4.97 ± 0.71a | 15.22 ± 0.43a | 5.20 ± 1.30a | 3.80 ± 0.90a | 22.40 ± 0.99a * |

Considering the last foliar application, a correlation of Pearson between the mineral elements analyzed (Ca, K, S, and P) (Figure 8) and the quality parameters (height, diameter, and dry weight content) (Table 1) in the tubers was carried out for both varieties (Tables 2 and 3).

**Table 2.** Pearson's correlation between Ca, K, S, P, height, diameter, and dry weight content in tubers after the last foliar application for the Picasso variety. Height (H); Diameter (D) and Dry weight (Dw).

|      | Ca     | K      | S      | P      | H      | D      | Dw     |
|------|--------|--------|--------|--------|--------|--------|--------|
| Ca   | 1      | −0.866 | −0.398 | −0.579 | 0.045  | 0.012  | 0.131  |
| K    | −0.866 | 1      | 0.515  | 0.603  | −0.040 | 0.022  | 0.045  |
| S    | −0.398 | 0.515  | 1      | 0.226  | 0.269  | −0.087 | 0.081  |
| P    | −0.579 | 0.603  | 0.226  | 1      | 0.245  | 0.125  | 0.059  |
| H    | 0.045  | −0.040 | 0.269  | 0.245  | 1      | −0.056 | −0.039 |
| D    | 0.012  | 0.022  | −0.087 | 0.125  | −0.056 | 1      | 0.817  |
| Dw   | 0.131  | 0.045  | 0.081  | 0.059  | −0.039 | 0.817  | 1      |

**Table 3.** Pearson's correlation between Ca, K, S, P, height, diameter, and dry weight content in tubers after the last foliar application for the Agria variety. Height (H); Diameter (D) and Dry weight (Dw).

|      | Ca     | K      | S      | P      | H      | D      | Dw     |
|------|--------|--------|--------|--------|--------|--------|--------|
| Ca   | 1      | −0.360 | −0.316 | −0.251 | −0.651 | −0.123 | −0.129 |
| K    | −0.360 | 1      | 0.970  | 0.955  | 0.273  | −0.637 | −0.532 |
| S    | −0.316 | 0.970  | 1      | 0.994  | 0.227  | −0.662 | −0.561 |
| P    | −0.251 | 0.955  | 0.994  | 1      | 0.162  | −0.685 | −0.585 |
| H    | −0.651 | 0.273  | 0.227  | 0.162  | 1      | −0.144 | −0.156 |
| D    | −0.123 | −0.637 | −0.662 | −0.685 | −0.144 | 1      | 0.974  |
| Dw   | −0.129 | −0.521 | −0.561 | −0.585 | −0.156 | 0.974  | 1      |

## 4. Discussion

The accumulation of minerals is mainly obtained from the soil solution by the roots [33] and potato plants are usually classified as inefficient regarding the acquisition of nutrients due to the poor root system (being shallow and less extended) [34]. In the phloem tissue, potassium and phosphorus are very mobile minerals; however, calcium is considered less mobile [33], as well as sulfur. As such, with Ca being almost immobile in the phloem, tubers rely on its delivery through the xylem [35]. However, foliar sprays with Ca can complement the xylem mass flow of Ca through phloem redistribution [32].

This study showed that despite some toxicity symptoms in both products (CaCl$_2$ and Ca-EDTA), the Ca content generally increased in the *S. tuberosum* L. organs of both varieties and that Ca, K, S, and P showed different accumulation patterns during the biofortification process. Considering the increase in Ca content, the experiment also showed that despite Ca being almost immobile in the phloem [35], foliar sprays with Ca can complement the xylem mass flow of Ca through phloem redistribution [32]. As such, considering that the accumulation of minerals is mainly obtained from the soil solution by the roots [33] and potato plants are usually classified as inefficient regarding the acquisition of nutrients due to the poor root system (being shallow and less extended) [34], foliar applications can be a strategy to increase one or more mineral elements, namely, Ca [32,36–38], Zn [4,39–41], Se [42–44], Mn [39], Mg [36,37], I [3], and/or K [36]. Nevertheless, biofortification can be a simple and effective strategy [40] to increase Ca intake in diets dominated by potatoes. Thus, it is important to take into account that even in the same production conditions and environment, differences can occur between uptakes by the shoot Ca content being correlated with the cell wall chemistry and cation-binding capacity [35] and that tuber genotypic variation can have an influence [45]. As such, probably because of these factors, there are different patterns of Ca accumulation in the different organs comparing both varieties (Figures 5–8).

Moreover, regarding both the orthophotomap and the NDVI model of both fields (Figure 2), overall, there is a good vigor for all the treatments except for 12 kg·ha$^{-1}$ Ca-EDTA (in Picasso and Agria) and 24 kg·ha$^{-1}$ CaCl$_2$ (in Agria), where chlorosis and necrosis on the tips of the leaves were observed. In fact, the negative effects of Ca-EDTA when applied repeatedly in plants were already being reported in tomato plants [46], and the

symptoms associated with chloride toxicity were also being reported in *S. tuberosum* L. [47]. Additionally, Ca-EDTA toxicity symptoms can be due to the fact that treatments with EDTA have higher loads of Na or Cl [46], considering that potato plants are known as Na-sensitive plants and Cl-nonsensitive plants [47].

Nevertheless, after the last two foliar applications, the highest Ca content in the roots (Figure 5) was obtained with Ca-EDTA treatments. Thus, in tubers after the 7FA (Figure 7), the highest Ca content was obtained in 24 kg·ha$^{-1}$ CaCl$_2$ (in Agria) and 24 kg·ha$^{-1}$ Ca-EDTA (in Picasso). As such, as seen in the roots and tubers, the increase in Ca accumulation in both organs indicates that Ca is redistributed through the phloem [32] despite being identified as relatively immobile [33,35]. Compared to the remaining mineral elements analyzed (Figure 8), tubers showed the lowest Ca content, being attributed to Ca transport being mainly via the xylem and due to tubers hardly transpiring [48]. Moreover, despite the negative effects of EDTA and chloride [46,47], the highest Ca content was shown in tubers almost ready for harvest (Figure 7).

Additionally, as seen also in our data, biofortification can lead to interactions among different nutrients [5–8] in different organs and stages of plant development. In fact, generally, Ca-EDTA treatments showed a higher content of Ca in the Picasso variety organs, while in the Agria variety, there is not a clear tendency regarding both products. Regarding the leaves (Figure 6), the Ca content was higher in all treatments sprayed with Ca regarding the control, accordingly to a previous study carried out also on *S. tuberosum* L. [37].

Moreover, during the biofortification process (Figure 8), K has the highest concentration in potato tubers, considering its central role in establish tubers and starch [48], plays an important role in the translocation of carbohydrates from leaves to tubers [49], tuber composition and tuber bulking being dependent on K nutrition [50]. In fact, due to the high need for K, the potato crop is used as an indicator crop for K$^+$ availability [19].

Additionally, K has a central role in leaf area development and photosynthesis [48], showing the highest content after Ca in the leaves (Figure 7). Nevertheless, foliar application of Ca had generally an insignificant effect on the K content in the leaves, and this is probably due to the antagonistic effect of Ca on K absorption by roots at higher levels [37]. Moreover, K shows a decrease with the development of the plants (being more evident in the Agria variety), in accordance with a study carried out in potato leaves [51].

Indeed, as previously reported in a study carried out in potato plants, the organs which contain more K than tubers are the vegetative part (stems and leaves) [52], considering that K is readily redistributed within the plant after being acquired by the plant roots and delivered to the shoots through the xylem [53].

Regarding P (Figures 5–8), the content in the different organs during the biofortification process and plant development do not vary much within the variety, probably due to P (as Pi) being able to move in both the xylem and phloem, having a constant loading and unloading into the different organs of the plant [54]. Moreover, for the growing of tubers, the main source of P is accumulated in the aerial parts of the plant, being crucial for the rate of tuber expansion [55]. Additionally, as previously seen in different potato varieties, the P content in leaves is always higher than the P content in tubers [56], in accordance with the data obtained for both varieties (Figure 8). Indeed, regarding leaves, there is a decrease in content during the production cycle in both varieties, already reported by a previous study carried out with potatoes [55]. Hence, P accumulates in tubers until the tubers achieve full maturity [56], in accordance with the data obtained for Picasso and in disagreement with the data obtained for Agria (showing a decrease in P accumulation during the production cycle) (Figure 8). This difference in accumulation regarding both varieties can be due to the variation among the genotypes [56]. Additionally, according to [55], other organs of *S. tuberosum* L., namely, the roots, can contribute P supply to the growing tubers. This could explain the decrease in P content in the roots of the Agria variety during the production cycle (Figure 5).

Roots (Figure 5) and leaves (Figure 7) showed the highest S content during the biofortification process, due to the highest amount of S being absorbed by the roots and

atmospheric S being absorbed by the higher plant (as SO$_2$) [48]. In general, the S content showed a tendency of leaves > roots > stems > tubers, not in accordance with other studies carried out [57]. In this framework, the different patterns of accumulation in both varieties during the biofortification process and plant development can be due to the remobilization of mineral nutrients during the different stages of plant development [58] and the different genotypes. In fact, there are differences in mineral contents among distinct genotypes of potato [59].

Additionally, the monitorization of dry matter content was carried out throughout the biofortification process, considering that is one of the characteristics that determine the texture of the potato tubers and a criterion for the classification of quality [60]. Throughout the experiment, no significant differences were observed in the dry matter content, regardless of the varieties, types of fertilizers, and fertilization rates used (Table 1). Indeed, only Agria was suitable for industrial processing after the last foliar application, having more than 20% of dry matter content [60]. Regarding height and diameter (Table 1), there was also oscillation in those quality parameters; however, according to Portuguese law [61] (the tuber's caliber needs to be higher than 3.5 cm), both varieties after the last foliar application are suitable for industrial processing.

Furthermore, regarding Tables 2 and 3, Pearson's correlation analysis showed that there is a better correlation (>0.5) between K and P, K and S, and between diameter and height in the Picasso variety, and between K and S, K and P, S and P, and diameter and dry weight in the Agria variety, according to [62]. As such, in both varieties, there was always a higher correlation between K and S, and K and P.

## 5. Conclusions

Through the monitorization of mineral content in *Solanum tuberosum* L. organs during Ca biofortification, the Ca content increases generally in the tubers, roots, stems, and leaves of both varieties despite the negative effects observed due to Ca-EDTA and CaCl$_2$ foliar applications. In tubers after the last foliar application, the highest Ca content was obtained with CaCl$_2$ for the Agria variety and with Ca-EDTA for the Picasso variety, showing, compared to the control, increases of 33% and 95.6%, respectively. The concentration of Ca seems to have a tendency to decrease from the aerial part to the tubers (leaves > stems > roots > tubers). Regarding the K, P, and S concentrations, there were different patterns of accumulation in both varieties. Tubers that were exposed to different concentrations of CaCl2 or Ca-EDTA, in terms of diameter, height, and dry weight content, were not significantly different, meaning that these tubers can be safety consumed by humans and the application of these products did not affect the quality. The process of biofortification does not affect the physiological metabolism, and after the last foliar application, only the Agria variety was suitable for industrial processing regarding the dry weight content. Furthermore, it can be concluded that this itinerary can be implemented by farmers and companies involved in potato production in Portugal.

**Author Contributions:** Conceptualization, A.R.F.C.; methodology, F.C.L., M.G.B., J.C.K., M.S., M.F.P. and F.H.R.; software, A.R.F.C.; formal analysis, A.R.F.C.; investigation, A.R.F.C., D.D., C.C.P., A.C.M. and I.C.L.; resources: J.C.R., J.M.N.S., M.M.S., I.P.P., P.L., P.S.-C., F.C.L., F.H.R., M.F.P. and M.S.; writing—original draft preparation, A.R.F.C. and F.H.R.; writing—review and editing, A.R.F.C. and F.H.R.; supervision, F.C.L.; project administration, F.C.L.; funding acquisition, F.C.L. and JCR.. All authors have read and agreed to the published version of the manuscript.

**Funding:** This work received funding from PDR2020-101-030719 and the Fundação para a Ciência e a Tecnologia, I.P. (FCT), Portugal, through the research units UIDP/04035/2020 (GeoBioTec), UIDB/00239/2020 (CEF), and UID/FIS/04559/2013 (LIBPhys). This work was further supported by the grant of the Fundação para a Ciência e Tecnologia (FCT) UI/BD/150806/2020.

**Institutional Review Board Statement:** Not applicable.

**Informed Consent Statement:** Not applicable.

**Data Availability Statement:** Not applicable.

**Acknowledgments:** The authors are grateful to Louricoop—Cooperativa de Apoio e Serviços do Concelho da Lourinhã—CRL—Portugal for technical assistance in the field's production.

**Conflicts of Interest:** The authors declare no conflict of interest.

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
