# Peer review of "Mineral Monitorization in Different Tissues of Solanum tuberosum L. during Calcium Biofortification Process"

_horticulturae, doi:10.3390/horticulturae8111020_

Round 1

Reviewer 1 Report

Dear authors, 

Please open the attached word files where the comments and inquiries need to reply.

Author Response

Concerning to reviewer 1:

Indications of the reviewer: “Keywords should be in alphabetical order and should not duplicate words appearing in the title”

Reply of the authors: The keywords were in alphabetical order. The authors removed the Solanum tuberosum L. considering that was the only duplicated word that also appears in the title of the paper.

Indications of the reviewer: “The author should spell out the following abbreviation “UVA“once set or mentioned.”

Reply of the authors: The authors thanks to the reviewer. The abbreviation was spelled out in the begging of the manuscript.

Indications of the reviewer: “Page 4 line 179. Left space between number and degree of Celsius “60oC”

Reply of the authors: The authors put the missing space between the number and degrees Celsius.

Indications of the reviewer: “Page 1 line 36: change the word “quality analyzed” to “quality analysis”

Reply of the authors: The authors corrected the mistake from “quality analyzed” to “quality analysis”

Indications of the reviewer: “In the statistical section, the authors should mention the kind of design that has been used (RCD or RCBD), how many replicates has been used in the experiment, and the software or the program that has been used to analyze the data.”

Reply of the authors:  The information about the replicated that has been used In the experiment and the software was added to the section 2.2..

Indications of the reviewer: “The authors should justify the figure well. Because they are not clear enough”

Reply of the authors:  The figure was justified.

Indications of the reviewer: “The author should revise the English grammar of the manuscript.”

Reply of the authors: The authors revised the English grammar throughout the manuscript.

Indications of the reviewer: “I suggest the author make Pearson’s correlation or any type of correlation among the endogenous calcium in plants and tested parameters to understand well the effect of Calcium fertilizers on growth parameters, yield, and other nutrients (N, P, K, and S).”

Reply of the authors: Considering that the harvest took place a few days after the 7th foliar application and yield data were only obtained at harvest, taking into account its great temporal proximity. The authors performed, as suggested by the reviewer, a Pearson correlation for the analyzed mineral elements (Ca, P, K and S) and the quality parameters (height, diameter and dry weight).

Indications of the reviewer: ” The authors should cover in the discussion section, the effect of Ca-EDTA toxicity on reactive oxygen species (ROS) production.10- The authors should add the further study in the conclusion section.”

Reply of the authors: The ROS may or may not be beneficial, depending on the concentration (Tripathi et al.,2020). However, as ROS were no measured, we cannot discuss their effect, despite being a metabolic product that regulate plant growth and development, functioning as a signal components in abiotic and biotic stress-related events (Mhamdi et al., 2018).

Tripathi, D., Nam, A., Oldenburg, D. J., & Bendich, A. J. (2020). Reactive oxygen species, antioxidant agents, and DNA damage in developing maize mitochondria and plastids. Frontiers in plant science, 11, 596.

Mhamdi, A., & Van Breusegem, F. (2018). Reactive oxygen species in plant development. Development, 145(15), dev164376.

Reviewer 2 Report

Comments to the Author:

In the article of “Mineral monitorization in different tissues of Solanum tuberosum L. during Calcium biofortification process”. The authors studied the treatment of plants using two concentrations and two types of solutions during calcium biofortification. The review is more interesting, in my opinion, just a few changes are required.

Specific recommendations:

1. Notice the units that appear in the full text. For example, line32 (12 and 24 kg ha-1) line133 (12 and 24 kg∙ha-1). Please check the full text for all similar situations to ensure consistency of the organization.

2. Some of the pictures in the article are inconsistent, can you make some adjustments? For example, in Figure 3 and Figure 4, in the case of the same number of pictures, try to keep the image size consistent.

3. line 209, why is there two lettersDmarked in the D chart?

4. Note that the abbreviations for the full text use the same form. For example,line 34S. tuberosum L.line 138S. tuberosum.

5. Please note that it is written correctly. Line 55KIO3.

6. Check that the specialist vocabulary that appears for the first time in the article has a full name. Line167 “NDVI”.

7. Some of the sentences in the article are not smooth, and it is recommended that the author improve it after reading through the article to improve the quality of the language.

8. In terms of conclusion, the summary of the article is not full enough, can it be more innovative?

9. This is a tedious introductory part, can it be compressed? It is recommended that you summarize the content of your predecessors, identify the existing problems, and clarify the necessity of your research.

10. Line 149,Changefig.1to “figure.1”. Please check the full text for a similar situation.

11. Authors should supplement the content of the article in the context of the most recently published literature. It is recommended to highlight the novelty of your research content in combination with the newly published articles in recent years.

12. The scale positions of figure 2 C& D are not the same. Please try to align them.

13. The text in Figure 7 & 8 is not the same size. Please adjust it.

14. Line56, “2.0 kg I ha-1”. Please note that it is written correctly.

15. Please adjust the position of the scale in Figure 2 to match.

16. Notice the units that appear in the full text. For example, line137 (4th, 6th, and 7th) line178 (4th, 6th and 7th). Please check the full text for all similar situations to ensure consistency of the organization.

17. Some of the images in the Figure and their subheadings are not aligned, as shown in Figure 3. Please adjust them to align.

Author Response

Concerning to reviewer 2:

Indications of the reviewer: ”Notice the units that appear in the full text. For example, line32 (12 and 24 kg ha-1) line133 (12 and 24 kg∙ha-1). Please check the full text for all similar situations to ensure consistency of the organization.

Reply of the authors: The authors uniformized the units in the full text.

Indications of the reviewer: ”Some of the pictures in the article are inconsistent, can you make some adjustments? For example, in Figure 3 and Figure 4, in the case of the same number of pictures, try to keep the image size consistent.

Reply of the authors: Both of the figures were adjusted, and the size were uniformized.

Indications of the reviewer: ”line 209, why is there two letters“D”marked in the D chart?

Reply of the authors: The extra letter D was eliminated by the authors, was a mistake of formatting.

Indications of the reviewer: ”Note that the abbreviations for the full text use the same form. For example,line 34“S. tuberosum L.”line 138“S. tuberosum”.

Reply of the authors: The authors uniformized the abbreviation throughout the manuscript.

Indications of the reviewer: ”Please note that it is written correctly. Line 55“KIO3”.

Reply of the authors: The authors corrected the “KIO3” to “KIO3”.

Indications of the reviewer: ”Check that the specialist vocabulary that appears for the first time in the article has a full name. Line167 “NDVI”.

Reply of the authors:  The NDVI abbreviation was explained in line 167.

Indications of the reviewer: ”Some of the sentences in the article are not smooth, and it is recommended that the author improve it after reading through the article to improve the quality of the language.

Reply of the authors: The authors carried out some changes though the manuscript to improve the quality of the language.

Indications of the reviewer: ”In terms of conclusion, the summary of the article is not full enough, can it be more innovative?

Reply of the authors: The authors remade the conclusion part of the manuscript.

Indications of the reviewer: ”This is a tedious introductory part, can it be compressed? It is recommended that you summarize the content of your predecessors, identify the existing problems, and clarify the necessity of your research.

Reply of the authors: The authors tried to write a well-documented introduction to explain the state-of-the-art of the subject and the reasons for the research and we kindly disagree with the reviewer. Additionally, other reviewers made very positive comments about the introduction.

Indications of the reviewer: ”Line 149,Change“fig.1”to “figure.1”. Please check the full text for a similar situation.

Reply of the authors: The change from “Fig.” to “Figure” was carried out though the manuscript

Indications of the reviewer: ”Authors should supplement the content of the article in the context of the most recently published literature. It is recommended to highlight the novelty of your research content in combination with the newly published articles in recent years.”

Reply of the authors: There isn’t much published research on potato biofortification via foliar applications, especially on Ca. In lines 418 and 419, recent studies on the subject carried out in potatoes are mentioned.

Indications of the reviewer: ”The scale positions of figure 2 C& D are not the same. Please try to align them.”

Reply of the authors: The authors scale positions of figure 2 C and D are the same.

Indications of the reviewer: ”The text in Figure 7 & 8 is not the same size. Please adjust it.”

Reply of the authors: The authors disagree with the reviewer, legend in Figure 7 and 8 is the same size (Palatino Linotype, 9).

Indications of the reviewer: ”Line56, “2.0 kg I ha-1”. Please note that it is written correctly.”

Reply of the authors: Its written correctly, and according to the cited paper.

Indications of the reviewer: ”Please adjust the position of the scale in Figure 2 to match.”

Reply of the authors: The figure 2 (A,B,C,D) have a specific scale, that is which is shown in each image.

Indications of the reviewer: ”Notice the units that appear in the full text. For example, line137 (4th, 6th, and 7th) line178 (4th, 6th and 7th). Please check the full text for all similar situations to ensure consistency of the organization.”

Reply of the authors: The authors unamortized to 4th, 6th and 7th, though the manuscript.

Indications of the reviewer: ”Some of the images in the Figure and their subheadings are not aligned, as shown in Figure 3. Please adjust them to align.

Reply of the authors: The imagens in the figure were adjusted and aligned.

Reviewer 3 Report

English editing of the entire manuscript is required.
Include reference for the Zero Hunger and the Sustainable Development Goals (line 47).

Author Response

Concerning to reviewer 3:

Indications of the reviewer: ”English editing of the entire manuscript is required”

Reply of the authors: The authors performed an English editing though the manuscript.

Indications of the reviewer: ”Include references for the Zero Hunger and the Sustainable Development Goals (line 47).”

Reply of the authors: The authors included the reference for the Zero Hunger and the Sustainable Development Goals in the manuscript.

Reviewer 4 Report

Comments and suggestions are marked in the attached file

Author Response

Concerning to reviewer 4:

Indications of the reviewer: ”Relate this process to the role of calcium in tissue/organ development”

Reply of the authors: The authors change the sentence to be in agreement with the information transmitted throughout the manuscript.

Indications of the reviewer: ”Tuber is not stem?”

Reply of the authors: The reviewer is right, in fact tubers are stem tubers, that grows underground through stolons. However, for a better understanding of the different organs into which the Solanum tuberosum L. plant was divided, the authors agreed to divide the organs into 4 parts (underground part – tubers and roots, and aerial part – stems and leaves).

Indications of the reviewer: ”Be careful with this statement, as plants do not show calcium toxicity, since this element needs to be very well regulated in the cytoplasm so as not to cause metabolic disorders. Review the way of writing. I imagine that this reported toxicity could be related to the form of application and the time of contact with the leaf or it could have induced a potassium deficiency, but not calcium toxicity.”

Reply of the authors:  The reviewer is absolutely right and the “toxicity” part was rewritten. The negative effects observe in Solanum tuberosum L. plants with the products applied (Ca-EDTA and CaCl2) are due to their composition. EDTA products have a higher load of Na or Cl and as stated in lines 432 – 438, potato plants are Na sensitive.

Indications of the reviewer: ”Here in the introduction, we will discuss works that report this phloem contribution in the transport of calcium to the tubercles and how this calcium is stabilized in the phloem, since this element cannot remain in its ionic form in the cytoplasm for a long time.”

Reply of the authors: The authors agreed with the reviewer.

Indications of the reviewer: ”How were the applications made?

Did you use a sprayer with CO2 pressure control?

What pressure is applied?

Did you use adjuvant?

What is the pH of the solution?”

Reply of the authors: The foliar application of the products was carried out using a manual sprayer, using the same application pressor as when applying plant phytopharmaceuticals products. The information about the pH of the solution was added to the section 2.1. as well as information about the dates of applications.

Indications of the reviewer: ”What is the (n) sample and the experimental design?”

Reply of the authors: The authors complemented that information in the 2.2. section.

Indications of the reviewer: ”I recommend rethinking the way to present the results. since I believe it is not a case of toxicity due to the application of calcium, but a deficiency of another element, perhaps potassium.

Even because the control was similar.”

Reply of the authors: The authors would like to know if the reviewer perhaps was not mistaken and wanted to refer to sodium instead of potassium. Despite there is a decrease in K content though the foliar applications, K values are within normal ranges (above 10000 ppm).

Indications of the reviewer: ”Note leaf K concentrations at these application concentrations.

Remember that Ca is immobile, as the application was foliar, it will remain on the leaf, which may interfere with the CAX carrier.”

Reply of the authors: Despite Ca is almost immobile in the phloem, foliar sprays can complement the xylem mass flow of Ca ( see lines 409 – 411).

Indications of the reviewer: ”Restructure the sentence! it is unclear”

Reply of the authors: The authors restructure the sentence.

Indications of the reviewer: ”Sulfur is immobile!!!!”

Reply of the authors: The authors corrected the mistake. Sulfur has poor mobility in plants.

Indications of the reviewer: ”Reconstruct the conclusion so that it relates to your goals and without repeating the results”

Reply of the authors: The conclusion was reconstructed by the authors.

Reviewer 5 Report

Potatoes are staple available food for large populations all over the world but not a very nutritive one; improving tubers quality is an important research for food security in disadvantaged areas or communities, and not only (everybody likes french fries, don't they?).

I think ”NDVI” acronym has to be explained

Well documented Introduction to explain the state-of-the-art of the subject and reasons for the research.

I don't very well understand how the CaCl2 and Ca-EDTA applications were carried out (rows 148-150 and 154-156). Seven applications in one day? Only one foliar application with Ca-EDTA 24 kg/ha but then other foliar applications? Maybe the authors could draw a fertilization scheme. And specify the date and/or vegetation phase when each application was made.

Figures and the Table are clear in presenting treatments application influence on plants mineral composition and morphologic parameters.

Very detailed and well explanatory discussions.

In row 464 what does ”Pi” mean?

Many, appropriate, and fairly new references

Author Response

Concerning to reviewer 5:

Indications of the reviewer: ”I think ”NDVI” acronym has to be explained”

Reply of the authors:  The authors explained in section 2.2. the acronym for NDVI.

Indications of the reviewer: ”Well documented Introduction to explain the state-of-the-art of the subject and reasons for the research.

I don't very well understand how the CaCl2 and Ca-EDTA applications were carried out (rows 148-150 and 154-156). Seven applications in one day? Only one foliar application with Ca-EDTA 24 kg/ha but then other foliar applications? Maybe the authors could draw a fertilization scheme. And specify the date and/or vegetation phase when each application was made.”

Reply of the authors: The foliar applications were carried out in a total of seven times, after the beginning of tuberization of both varieties, in the dates presented in section 2.1.. However, due to the negative effects observed right after the first foliar application of Ca-EDTA 24 kg/ha, it was decided no to carry out any further foliar applications. Thus, instead of 7 foliar applications with Ca-DTA 24 kg/ha, only a single application was carried out on 30 May. The remain treatments (CaCl2 12 kg/ha, CaCl2 24 kg/ha and Ca-EDTA 12 kg/ha) were performed normally at 30 May, 7 June, 14 June, 21 June, 28 June, 4 July and 12 July.

Indications of the reviewer: ”Figures and the Table are clear in presenting treatments application influence on plants mineral composition and morphologic parameters”

Reply of the authors: The authors thanks to the reviewer.

Indications of the reviewer: ”Very detailed and well explanatory discussions”

Reply of the authors: The authors thanks to the reviewer.

Indications of the reviewer: ”In row 464 what does ”Pi” mean?”

Reply of the authors: “Pi” is the most readily form of P accessed by plants in soil solution, according to Schachtman et al. (1998).

Schachtman, D. P., Reid, R. J., & Ayling, S. M. (1998). Phosphorus uptake by plants: from soil to cell. Plant physiology, 116(2), 447-453.

Indications of the reviewer: ”Many, appropriate, and fairly new references”

Reply of the authors: The authors thanks to the reviewer.

Round 2

Reviewer 1 Report

The authors have made all the corrections. Therefore, I recommended publishing the manuscript after making the following modification to the format:

1- On page 7: Remove the space above Figure 4,

2- On page 5, The word "Results" should be in a separate line, 

3- the Axises of Y and X of all charts (Figure) should change from white color to black.

Author Response

After reading and considering the review perspectives of the paper “Mineral monitorization in different tissues of Solanum tuberosum L. during Calcium biofortification process”, the authors of the manuscript reply as follows:

  1. Concerning to Reviewer 1:

Indications of the reviewer: “1- On page 7: Remove the space above Figure 4,

Reply of the authors: The authors removed the space above Figure 4.

Indications of the reviewer: ” 2- On page 5, The word "Results" should be in a separate line, 

Reply of the authors: The authors made separate the word “Results” in other line.

Indications of the reviewer: ” 3- the Axises of Y and X of all charts (Figure) should change from white color to black.

Reply of the authors: The authors change the color of the Y and X axis in all the charts.

Reviewer 2 Report

1. Pay attention to the units appearing in the full-text, line 177:12 and 24kg ha-1, line 152:24kg ha-1

2. Line 197: put "3. Result" in the right place.

3. line204: Please check whether the "6th" format is correct.

4. Why choose "4, 6, and 7" for foliar fertilization?

5. line373: Missing comma after "(Table 1)"

6. Why did the author choose "12 kg · HA - 1" and "24 · kg ha - 1"?

Author Response

Reply to the review

After reading and considering the review perspectives of the paper “Mineral monitorization in different tissues of Solanum tuberosum L. during Calcium biofortification process”, the authors of the manuscript reply as follows:

Indications of the reviewer: ” 1. Pay attention to the units appearing in the full-text, line 177:12 and 24kg ha-1, line 152:24kg ha-1

Reply of the authors: The units are written in the same way “ 12 and 24 kg∙ha-1”, perhaps the reviewer didn’t see that are corrected through track changes through the manuscript.

Indications of the reviewer: ” 2. Line 197: put "3. Result" in the right place.

Reply of the authors: The Results are in the right place. Figures 2, 3 and 4 expressed results as well as the following figures and table.

Indications of the reviewer: ” 3. line204: Please check whether the "6th" format is correct.

Reply of the authors: The authors corrected “6th” to “6th”.

Indications of the reviewer: ” 4. Why choose "4, 6, and 7" for foliar fertilization?

Reply of the authors: In previous studies carried out by our research group*, we found that Solanum tuberosum L. plants after the 4th foliar applications didn’t show any negative effects - the harvest data occurred 30 days after the application. As such, we extended the number of foliar applications beyond the 4th one (5th 6th and 7th) and the harvest date took place 17 - 28 days after the 7th application, where the monitorization of mineral elements (with special attention to Ca) in the plant organs, took place.

*Coelho, A.R.F.; Lidon, F.C.; Pessoa, C.C.; Marques, A.C.; Luís, I.C.; Caleiro, J.; Simões, M.; Kullberg, J.; Legoinha, P.; Brito, M.; Guerra, M.; Leitão, R.G.; Galhano, C.; Scotti-Campos, P.; Semedo, J.N.; Silva, M.M.; Pais, I.P.; Silva, M.J.; Rodrigues, A.P.; Pessoa, M.F.; Ramalho, J.C.; Reboredo, F.H. Can Foliar Pulverization with CaCl2 and Ca(NO3)2 Trigger Ca Enrichment in Solanum tuberosum L. Tubers? Plants 202110, 245. https://doi.org/10.3390/plants10020245

Coelho, A.R.F.; Ramalho, J.C.; Lidon, F.C.; Marques, A.C.; Daccak, D.; Pessoa, C.C.; Luís, I.C.; Guerra, M.; Leitão, R.G.; Semedo, J.M.N.; Silva, M.M.; Pais, I.P.; Leal, N.; Galhano, C.; Rodrigues, A.P.; Legoinha, P.; Silva, M.J.; Simões, M.; Scotti Campos, P.; Pessoa, M.F.; Reboredo, F.H. Foliar Spraying of Solanum tuberosum L. with CaCl2 and Ca(NO3)2: Interactions with Nutrients Accumulation in Tubers. Plants 202211, 1725. https://doi.org/10.3390/plants11131725

Indications of the reviewer: ” 5. line373: Missing comma after "(Table 1)"

Reply of the authors: The authors thanks to the reviewer and corrected from “Table 1” to “Table 1.”.

Indications of the reviewer: ” 6. Why did the author choose "12 kg · HA - 1" and "24 · kg ha - 1"?

Reply of the authors: In previous research with 6 kg/ha (CaCl2) no negative effects were observed on potato plants. Thus, our option was to double the concentration used. As we verified that some authors make experiments with levels >20 kg/ha, we thought that the threshold could be increased until 24 kg/ha.

For instance, Ozgen et al. (2006) applied 21 kg/ha of CaCl2 in cv. Russet Burbank and Ca content in the tubers increased. Also, despite being other form of Ca application (Ca(NO3)2), seven applications with 20, 40, 60, 80, 100 or 120 kg/ha were carried out by Hamdi et al. (2015), showing a increased in tubers and leaves of Ca content.

More recently in fact, Loekito et al. (2022) carried out a study with pineapples, in which four foliar applications with Ca-EDTA and CaCl2 were carried out with a concentration of 75kg/ha, increasing the Ca content of the crop.

Ozgen, S.; B. H. Karlsson and J. E. Palta (2006). Response of potatoes (cv russet burbank) to supplemental calcium applications under field conditions: tuber calcium, yield and incidence of internal brown spot. Amer. J. of Potato Res., 83:195-204.

Hamdi, W.; L. Helali; R. Beji; K. Zhani; S. Ouertatani and A. Gharbi (2015). Effect of levels calcium nitrate addition on potatoes fertilizer. IRJET, 2(3): 2006-2013.

Loekito, S.; Afandi, A.; Nishimura, N.; Koyama, H., and Senge, M. (2022). The Effects of Calcium Fertilizer Sprays during Fruit Development Stage on Pineapple Fruit Quality under Humid Tropical Climate. International Journal of Agronomy2022.